# 🗺️ PointArena: Probing Multimodal Grounding Through Language-Guided Pointing

## Abstract

Pointing serves as a fundamental mechanism for grounding language within visual contexts, with applications spanning robotics, assistive technologies, and interactive AI systems. While recent multimodal models have begun supporting pointing capabilities, existing benchmarks typically focus only on referential object localization. We introduce **PointArena**, a comprehensive platform for evaluating multimodal pointing across diverse reasoning scenarios. PointArena comprises three components: (1) **Point-Bench**, a curated dataset of approximately 1,000 pointing tasks across five reasoning categories; (2) **Point-Battle**, an interactive web-based arena facilitating blind, pairwise model comparisons, which has collected over 4,500 anonymized votes; and (3) **Point-Act**, a real-world robotic manipulation system allowing users to directly evaluate model pointing in practical settings. We conducted extensive evaluations of state-of-the-art open-source and proprietary models. Results indicate that `Molmo-72B` consistently outperforms others, though proprietary models demonstrate comparable performance. Additionally, we find that supervised training targeting pointing tasks significantly improves performance. Across our multi-stage evaluation pipeline, we observe strong correlations, underscoring the role of precise pointing in enabling multimodal models to bridge abstract reasoning with real-world actions.

## 1 Introduction

Pointing focuses our attention. It is one of the earliest and most universal non-verbal methods we use to communicate intent; in fact, children learn to point as a prelinguistic form of communication (Tomasello et al., 2007). Precise spatial grounding—*pointing*—enables a wide range of practical and high-impact applications across robotics, assistive technology, human-computer interaction, and vision-language interfaces. In robotics, a pointing-capable model can interpret language commands like "pick up the red cup next to the bowl" and translate them into precise spatial actions (Yuan et al., 2025), enabling fine-grained object manipulation in cluttered environments (Duan et al., 2024). In assistive technologies, systems can help visually impaired users by answering spatial queries such as "where is the handle on this door?" (Bigham et al., 2010) or 'which one is the garlic?' In education or creative tools, pointing allows for interactive visual tutoring, such as identifying components in scientific diagrams or guiding a learner through a painting (Hu et al., 2024). Even in everyday virtual assistants or search engines, the ability to refer to specific image regions via pointing could make multimodal interactions more intuitive and expressive (Deitke et al., 2024a). Across these domains, pointing provides a low-bandwidth yet powerful spatial interface for grounding language in vision—precise enough for manipulation, intuitive enough for communication, and general enough to scale with multimodal models.

Recent advances in multimodal models have begun to incorporate more dynamic and spatially expressive forms of interaction. The Segment Anything Model (SAM) (Kirillov et al., 2023) enables segmentation from sparse visual prompts such as points or boxes, revealing the potential of fine-grained spatial control. Google's Gemini models (Georgiev et al., 2024) push the boundaries of long-context visual reasoning, incorporating

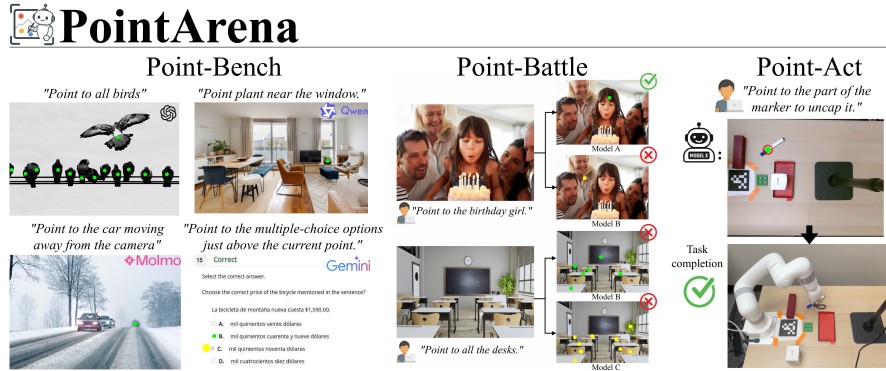

Figure 1: **Overview of *PointArena*.** PointArena consists of three components: **Point-Bench**, a curated dataset for evaluating grounded pointing across five reasoning types; **Point-Battle**, a live platform for blind, pairwise model comparisons with user voting; and **Point-Act**, real-world task involving manipulation via pointing-based language commands.

multiple modalities over extended sequences. In parallel, new datasets have emerged to support explicit spatial referencing. Molmo's PixMo dataset (Deitke et al., 2024a) introduces 2D pointing as a form of multimodal alignment between images and instructions, while RoboPoint (Yuan et al., 2025) focuses on spatial affordance prediction by linking instructions to interaction-relevant keypoints in robotic contexts. These setups bias evaluations toward pixel-level accuracy rather than conceptual reasoning and often lack diversity or scalability.

There is a need for a holistic evaluation platform to make progress towards language-guided pointing. Although datasets for referring expressions exist (e.g.RefCOCO, RefCOCO+, and RefCOCOg (Kazemzadeh et al., 2014; Yu et al., 2016a)), they are focused on a subset of pointing tasks: object location. They lack the ambiguity and contextual variability that users expect from modern interactive models, limiting their utility for studying pragmatic or interactive applications. As a result, they offer only partial insight into the full spectrum of grounding required for embodied or assistive agents. We, therefore, propose **PointArena**, a platform to probe and evaluate grounded visual reasoning with pointing. PointArena presents a suite of tasks where a multimodal model must answer questions or resolve instructions by combining language and pointing gestures to identify specific image regions. These tasks go beyond traditional VQA (Antol et al., 2015) by requiring spatial outputs (e.g., selecting a location or region) rather than purely textual ones. PointArena allows for both unambiguous and ambiguous scenarios, supporting studies of disambiguation, spatial commonsense, and pragmatic inference. Unlike bounding boxes, segmentation masks, or free-form text responses, pointing offers high-precision signal that avoids reliance on object contours or dense annotations and is directly compatible with human evaluation.

**PointArena** decomposes pointing into three stages of evaluation: 1) **Point-Bench** is a curated dataset of 982 manually selected, annotated, and verified image-question pairs across five high-level categories (Spatial, Affordance, Counting, Steerable, and Reasoning). 2) **Point-Battle** an interactive, online platform for blind, pairwise comparison between models based on user instructions. Users select from curated or custom-uploaded images. Voting is anonymized, and we have collected over $4,500$ votes from more than $100$ participants. 3) **Point-Act** is a real-world benchmark that evaluates the utility of pointing in for a downstream application. The system directs a robotic arm to manipulate objects through pointing-based language commands. All three evaluation stages require minimal human effort; each is self-contained and can run live to evaluate any model. Through our evaluation of both open-source and proprietary models across the

three stages of the PointArena benchmark, we find that `Molmo-72B` achieves the highest performance on Point-Bench, with proprietary models such as `Gemini-2.5-Pro` performing comparably. Models trained with explicit pointing supervision consistently outperform those without. We also observe a strong correlation between static benchmark accuracy and human preference in Point-Battle. Notably, we find that adding language reasoning (e.g., Chain-of-Thought (Wei et al., 2022)) does not improve visual grounding for pointing tasks. Our study further reveals several other actionable insights into model behavior and evaluation design. We see **PointArena** as a missing component necessary as we develop general-purpose vision-language models that can reason about and interact with the world.

## 2 RELATED WORK

**Grounding benchmarks.** Benchmarks for visual grounding and spatial reasoning span 2D and 3D. Re-fCOCO/RefCOCO+/RefCOCOg target 2D referring expressions, with RefCOCOg emphasizing longer, fine-grained descriptions (Yu et al., 2016b). In 3D, ScanRefer provides language–geometry supervision for object localization (Chen et al., 2020), while ReferIt3D and CityRefer extend to fine-grained and outdoor settings with geographic cues (Achlioptas et al., 2020; Miyanishi et al., 2023). Interactive datasets such as GuessWhat?! evaluate multi-turn object grounding (de Vries et al., 2017), and Flickr30K Entities supports phrase–region alignment (Plummer et al., 2016). Most focus on bounding boxes, retrieval, or dialog-based localization rather than explicit pointing behavior.

**Arena-style evaluation.** Arena evaluations compare models via anonymized pairwise votes, popularized by *Chatbot Arena* (Chiang et al., 2024). Extensions include multi-turn AI scoring (*MT-Bench*) (Zheng et al., 2023), rank stabilization (*am-ELO*) (Liu et al., 2025), and automated content/voting (*Auto-Arena*) (Zhao et al., 2024). Tools like *BenchBuilder*, *Werewolf Arena*, and *OpenArena* broaden domains and offline use (Bailis et al., 2024; SYV-AI, 2024). Despite scalability and user alignment, these methods lack ground-truth guarantees and can be gamed (Min et al., 2025), motivating alternatives like *Tournament Evaluation* (Kelley & Wilson, 2025). *PointArena* addresses this by pairing ground-truth evaluation (Point-Bench) with human-preference battles (Point-Battle).

**Models that point or sketch.** Recent MLLMs advance vision–language reasoning (e.g., GPT-4V) (Yin et al., 2024). Architectures often align visual features to LMs (e.g., MiniGPT-4 with Q-Formers) (Wang et al., 2024). Molmo directly regresses normalized coordinates, achieving strong 2D grounding (e.g., icon localization) (Deitke et al., 2024b). RoboPoint instruction-tunes VLMs for robotic affordance pointing, out-performing GPT-4o and PIVOT on spatial accuracy (Yuan et al., 2024; Nasiriany et al., 2024). Other systems add region-level control, multilinguality (VisCPM, Qwen-VL), or multi-modal inputs (NExT-GPT) (Yin et al., 2024). Nonetheless, precise, reliable spatial localization remains challenging, underscoring the need to study what enables effective pointing in MLLMs.

## 3 POINTARENA

Evaluating the ability of MLLMs to localize language-referred entities in images requires benchmarks that are both precise and diagnostic. Existing benchmarks often emphasize classification or captioning, but fall short when it comes to assessing fine-grained spatial grounding—the ability to resolve natural language instructions into specific image coordinates. This capability is critical not only for understanding model alignment with human intent, but also for enabling downstream applications in robotics (Yuan et al., 2024), augmented reality (Duan et al., 2023), and interactive web agents (Gur et al., 2023), and potentially contributing to explainability (Park et al., 2018). As both specialized pointing models and general-purpose MLLMs improve, standardized evaluation across open-source and proprietary systems becomes essential.

We introduce **PointArena**, an evaluation suite for language-conditioned pointing, comprising three stages: (i) **Point-Bench**, a curated dataset for controlled measurement of spatial localization accuracy; (ii) **Point-Battle**, a live, blinded human-preference arena for pairwise model comparison; and (iii) **Point-Act**, a real-world robotic setting that evaluates pointing precision through physical execution. Together, these components form a unified framework to quantify and analyze how well MLLMs ground language into action.

## 3.1 TASK FORMULATION

We formalize pointing as a language-conditioned fine-grained localization task: given an RGB image $I \in \mathbb{R}^{H \times W \times 3}$ and a natural-language prompt $q = \{w_t\}_{t=1}^{T}$, a multimodal large language model $\mathcal{F}_\theta$ maps $(I, q)$ to a set of image-space coordinates $P = \{(x_i, y_i)\}_{i=1}^{K}$ with $0 \le x_i \le W-1$ and $0 \le y_i \le H-1$. Ground-truth supervision is provided as binary masks $\{M_j\}_{j=1}^{K^*}$, where each $M_j \in \{0, 1\}^{H \times W}$ denotes the valid region for one of the $K^*$ annotated targets. A predicted point $(x_i, y_i)$ is correct if it falls within the spatial support of some mask, i.e., $\exists j : M_j[y_i, x_i] = 1$. We deem a prediction *successful* iff the cardinalities match ($K = K^*$) and every target region is covered, i.e., $\forall j \, \exists (x_i, y_i) \in P$ such that $M_j[y_i, x_i] = 1$. This formulation enables fully automated evaluation from masks alone, with no human required at test time.

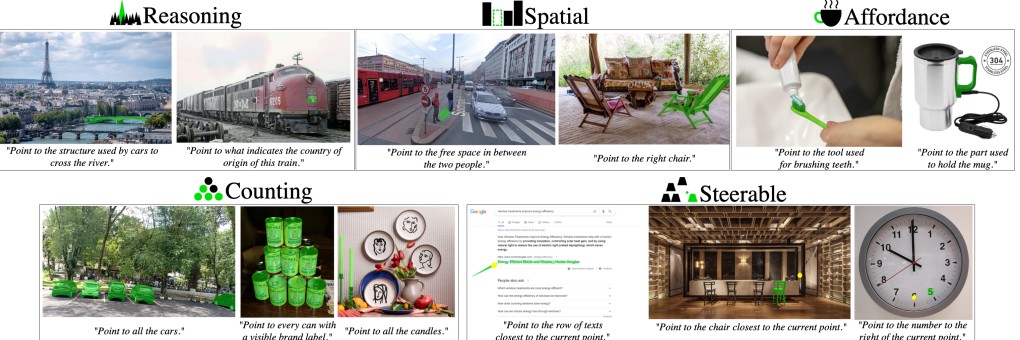

Figure 2: **Overview of the five Point-Bench categories and the annotation UI.** Point-Bench includes 982 image-query pairs grouped into five categories: **Spatial** (positional references), **Affordance** (functional part identification), **Counting** (attribute-based grouping), **Steerable** (relative pointing), and **Reasoning** (open-ended visual inference). Each example shows a representative query and the corresponding target. On the right, we show the Gradio-based annotation interface used to collect and refine segmentation masks. Initial masks are generated using SAM and refined by annotators, followed by manual verification.

## 3.2 POINT-BENCH

Point-Bench is the largest benchmark for evaluating language-guided pointing, comprising 982 text-image pairs with pixel-level target masks collected from public sources after April 20, 2025. The dataset is evenly divided into five task-driven categories—Spatial, Affordance, Counting, Steerable, and Reasoning—derived from a survey of question types frequently tackled by open-source MLLMs (Deitke et al., 2024b; Yuan et al., 2024; Team et al., 2025). Each category targets a distinct capability: 1) *Spatial* focuses on positional queries within scenes rich in spatial relationships or repeated objects (e.g., "Point to the leftmost tree in the image"); 2) *Affordance* emphasizes functional parts of objects, typically in tabletop scenes, prompting queries like "Point to the handle used for pouring"; 3) *Counting* features multiple similar items and supports queries about subsets based on number or attributes, such as "Point to all the blue cars in the image"; 4) *Steerable* leverages images from the PixMo dataset that include a reference point, guiding annotators to ask relative-position questions like "Point to the item closest to the marked point"; and 5) *Reasoning* presents event-rich or abstract

scenes, inviting open-ended queries that require inference, such as "Point to the tallest man-made object in the image." Annotators, recruited via crowdsourcing, were free to ask any question, but carefully selected category-specific images naturally guided them toward prompts aligned with each reasoning axis. These curated splits together support systematic evaluation of an MLLM's ability to recognize, reason, and precisely ground language in visual space.

To construct a Point-Bench, we developed an intuitive Gradio-based annotation interface. Annotators were shown images sampled from each category and asked to write natural language queries aligned with the category theme. These queries were then evaluated using predictions from three anonymized MLLMs. If one or fewer models produced a correct prediction as judged by human evaluators, the query was considered sufficiently challenging and accepted for inclusion in the dataset. Following this, the annotators used the same interface to annotate the target points directly on the image. A SAM model was used to generate initial masks based on the selected point, and users could refine these masks by editing or removing portions before submission. Finally, a separate group of annotators manually verified the masks to ensure they accurately reflected the user-generated queries.

### 3.3 POINT-BATTLE

As MLLMs increasingly incorporate visually grounded reasoning and pointing capabilities, static benchmarks become inadequate for evaluating performance in open-ended, real-world scenarios—particularly with respect to human preferences. To address this limitation, we introduce **Point-Battle**, a dynamic platform for pairwise evaluation of MLLMs' pointing abilities based on user-provided language instructions. Point-Battle adopts a head-to-head evaluation format inspired by Chatbot Arena (Chiang et al., 2024), implemented via a Gradio-based web interface. In each round, two anonymized models are randomly sampled from the top performers in **Point-Bench**—including GPT-4o, Gemini 2.5 Flash, Molmo-7B-D, Qwen2.5-VL-7B, and Grok-2 Vision. Users submit a natural language instruction and select an image from a curated dataset (post-April 20, 2025) or upload their own. The two models return point predictions, which are displayed side by side. Participants vote for the better output or select both good" or both bad" if applicable. No preset prompts are provided, encouraging diverse and unbiased instructions. Model identities are kept anonymous to prevent bias. Since launch, Point-Battle has collected over 4,500 votes from approximately 100 participants worldwide. Unlike the static **Point-Bench**, which may be subject to overfitting during model development, Point-Battle serves as a continuously updated benchmark that captures real-time human preferences and tracks progress in visually grounded reasoning across MLLMs. As Point-Battle scales, it also serves as a platform for collecting pointing data.

### 3.4 POINT-ACT

The first two stages of Point Arena evaluate MLLMs' pointing capabilities through quantitative metrics and human preference assessments. However, pointing is only meaningful insofar as it enables real-world utility. To evaluate such support, we introduce *Point-Act*—an interactive system where users issue natural-language instructions via a GUI to a double-blind MLLM. The model generates one or more predicted points, which are translated into actionable commands for an xArm 6 Lite robot as shown in Figure 7. The robot executes a pick-or-place action at the indicated location using depth sensing for spatial reasoning. This setup operationalizes pointing into end-to-end physical manipulation, bridging language grounding with robotic control. Point-Act highlights the downstream consequences of grounding precision: even small localization errors can cause execution failures, whereas accurate predictions enable consistent real-world success.

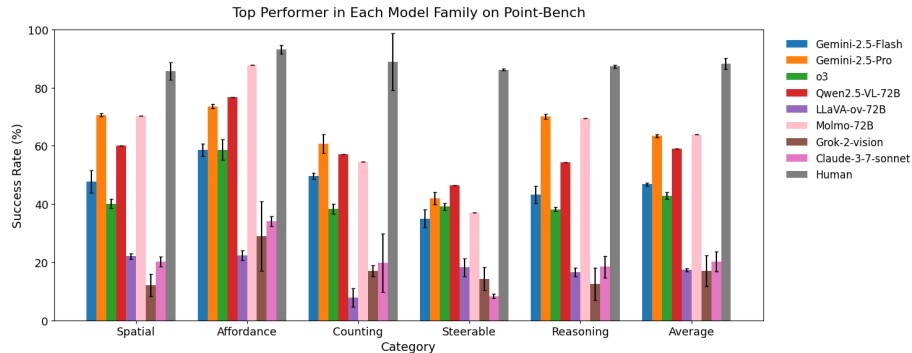

Figure 3: **Success rates of MLLMs on Point-Bench** across six task categories: *Spatial*, *Affordance*, *Counting*, *Steerable*, *Reasoning*, and *Average*. Each bar represents the mean success rate (%) for a given model, with error bars indicating standard deviation across three evaluation runs. The "Human" bar serves as an upper-bound reference. The results demonstrate substantial performance disparities, with top models (e.g., GPT-4o, Gemini-2.5-Pro, Molmo-72B) achieving near-human accuracy in select categories, while others (e.g., LLaVA, Grok, and Claude) consistently underperform.

## 4 EXPERIMENTS

We evaluate a range of multimodal large language models (MLLMs)—both proprietary and open-source—using three components: **Point-Bench** (static benchmark evaluation), **Point-Battle** (human preference comparison), and **Point-Act** (real-world robotic execution). Section 4.1 describes the evaluation protocols, including model selection, prompting, and success metrics. Section 4.2 presents results on model performance and the impact of pointing supervision. Section 4.3 presents results demonstrating the correlation between benchmark accuracy, human preference judgments, and real-world task performance in pointing tasks. Section 4.4 includes ablations on prompt structure and output formats using GPT-4o to analyze factors affecting pointing accuracy.

### 4.1 EVALUATION SETUP

All evaluations were performed under zero-shot prompting conditions. To ensure consistent outputs across models with differing internal coordinate systems—particularly proprietary ones—we adopted a standardized output format: $[x, y]$, where $x$ and $y$ denote horizontal and vertical pixel coordinates, respectively. This format was used across all models, except for those like `Molmo`, `Qwen2.5-VL`, and `Gemini`, which provide explicit coordinate outputs or prompting instructions. Success was measured using a binary metric: a prediction was considered correct if the point lay within the target mask. For non-counting tasks, models were prompted to predict a single point; if multiple were returned, only the first point was evaluated, assuming it reflected the highest-confidence prediction due to the autoregressive generation process.

**Point-Bench.** We benchmarked 16 MLLMs (spanning open-source and proprietary models, including key variants). Each model was evaluated on the same 982 image-instruction pairs, three times independently, to compute means and standard deviations. Open-source models were executed locally on NVIDIA A100 GPUs, while proprietary models were accessed via public APIs.

**Point-Battle.** To measure alignment with human preferences, we released a live evaluation platform and promoted it via social media and mailing lists. Users voted on head-to-head comparisons between anonymous

model outputs. Elo ratings were computed from pairwise comparisons excluding ambiguous votes ("both good" or "both bad").

**Point-Act.** We recruited 10 remote participants to interact with our real-world robot setup. For a fixed scene, participants evaluated three agents—`Molmo-7B-D`, `GPT-4o`, and a human reference—across three trials. After each condition, they completed a System Usability Scale (SUS) survey.

**Models.** We evaluate variants from Molmo (Deitke et al., 2024a), Gemini (Team et al., 2025), OpenAI (Achiam et al., 2023), Claude (Anthropic, 2024), Grok (xAI, 2024), LLaVA (Li et al., 2024), and Qwen (Bai et al., 2025). See appendix for details.

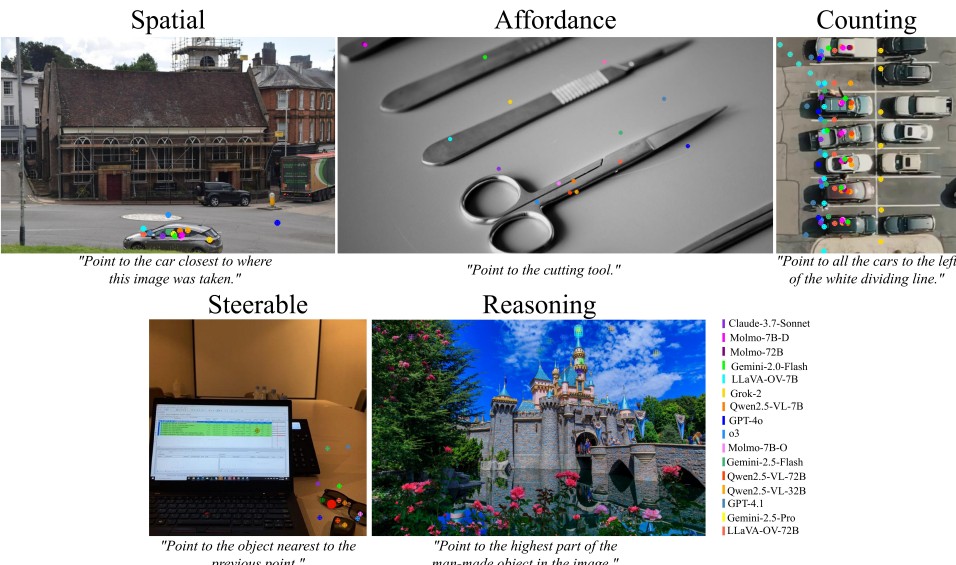

Figure 4: **Qualitative predictions across Point-Bench categories.** Example model predictions are shown for each of the five Point-Bench categories: **Spatial**, **Affordance**, **Counting**, **Steerable**, and **Reasoning**. Each colored dot corresponds to a prediction from a different MLLM, labeled by model name in the legend. These examples highlight the diversity of pointing behaviors and the variation in performance across models.

**Open-source models perform comparably to proprietary models in pointing accuracy.** Point-Bench results show that open-source MLLMs explicitly trained on pointing data often match or outperform proprietary models. For example, `Molmo-72B` outperformed `Gemini-2.5-Pro` by 0.43 percentage points—a statistically insignificant margin ($p \approx 0.29$). In affordance reasoning, open-source models like `Molmo-72B` and `Qwen2.5-VL` consistently exceed proprietary baselines. Overall, Molmo-72B achieves the highest performance on the Point-Bench benchmark as shown in Table 4.

**Pointing supervision significantly boosts performance.** Access to explicit pointing data is a key driver of model accuracy as shown in Figure 5a. Within the Qwen family, incorporating the PixMo corpus into `Qwen2.5-VL-7B` increased performance to 52.3%, a substantial gain over the 17.4% achieved by `Qwen2-VL-7B`, which did not use such data. In contrast, LLaVA variants—also trained without explicit pointing supervision—achieved only 4.8–17.4% on average.

**Proprietary models likely benefit from open-source pointing datasets.** While proprietary training data is opaque, we observe large performance jumps in models released shortly after PixMo (Deitke et al., 2024b) and RoboPoint (Yuan et al., 2024). For instance, `GPT-o3` improved by 21.1 percentage points

over `GPT-4-Turbo`, and `Gemini-2.5-Flash` improved by 45.9 points over `Gemini-1.5-Flash` (Figure 5a). These results suggest that proprietary models may have incorporated PixMo or a similar corpus.

**Open-source models align more closely with human preferences.** In Point-Battle, `Molmo-7B-D` outperformed `Gemini-2.5-Flash` by 196 Elo points. Their 95% confidence intervals do not overlap, and `Molmo-7B-D` won 79% of the 115 direct head-to-head comparisons, as shown in Figure 6. Both `Qwen2.5-VL-7B` and `Molmo-7B-D` surpass proprietary models in human preference evaluations and exceed the 1000-point baseline, indicating a statistically significant advantage over random guessing. However, in terms of preference-aligned pointing performance, `Molmo-7B-D` remains clearly superior to `Qwen2.5-VL-7B`.

**Molmo excels on Point-Act evaluation.** User study results shown in Figure 8 that `Molmo-7B-D` outperforms the proprietary `GPT-4o` model by a substantial margin, achieving 65% higher performance and approaching human (oracle) baseline levels. This superiority is also reflected in user preference, with `Molmo-7B-D` scoring 60.3 points higher in SUS than `GPT-4o`.

### 4.2 MAIN RESULTS

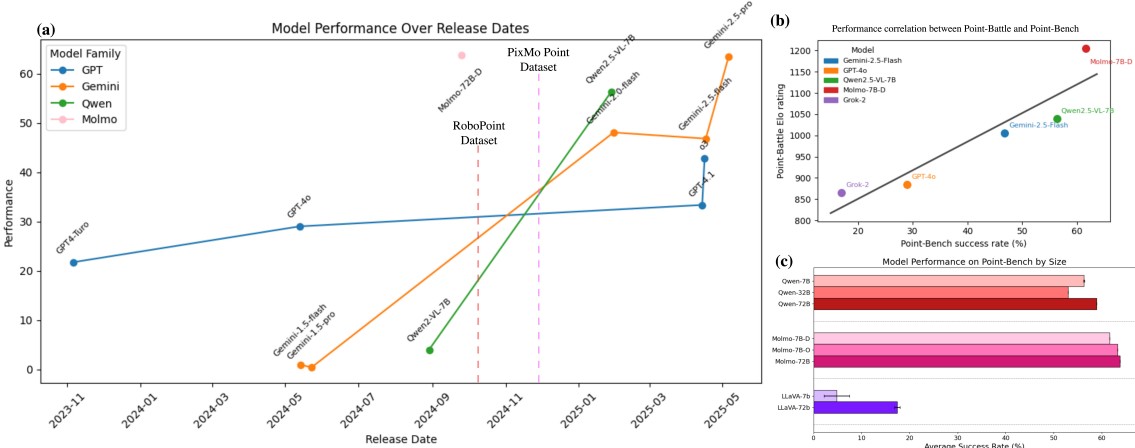

Figure 5: **Insights from Point-Battle and Point-Bench.** (a) Point-Bench accuracy over time by model family. Performance jumps notably post-PixMo (dashed line, Dec 2024) or post-RoboPoint dataset(Oct 2024), with `GPT-4.1` and `Gemini-2.0-Flash` showing large gains over predecessors, hinting at pointing supervision. (b) Point-Battle and Bench are strongly correlated ($R^2 = 0.85$), validating consistency. (c) Open-source model performance vs. size shows only minor gains, indicating diminishing returns with scale.

**Model size does not impact pointing performance.** As shown in Figure 5c, the performance of open-source models (LLaVA-OV, Molmo, and Qwen-VL) on Point-Bench remains largely unchanged with increased model size. For example, `Qwen2.5-VL-7B` performs within 3% of `Qwen2.5-VL-72B`, and `Molmo-7B-O` differs by less than 1% from `Molmo-72B`. These results suggest that scaling model size does not significantly improve pointing accuracy.

### 4.3 RESULTS BETWEEN THREE EVALUATION FRAMEWORKS.

The three-stage evaluation of MLLMs' pointing capabilities should not be viewed as isolated components, but as complementary steps in a progressive pipeline. As MLLMs improve, they are expected to advance

through these stages. Therefore, understanding the correlation and agreement between stages is crucial for assessing consistent performance gains.

**Human-preference and static dataset evaluations are highly consistent.** Point-Bench's static dataset will inevitably plateau as MLLMs improve by training on ever-larger, real or synthetic pointing corpora (e.g., RoboPoint (Yuan et al., 2024)). To stay ahead, we introduce **Point-Battle**, a live arena that updates continuously and enables open-ended model comparison in real time. Validating this setup, we re-evaluated the models tested on Point-Bench and observed strong alignment: Point-Battle scores correlate with Point-Bench results at $R^2 = 0.85$ (Figure 5b).

**Point-Bench accuracy predicts real-world task success.** We validated Point-Bench as a reliable proxy by testing three agents—`Molmo-7B-D`, `GPT-4o`, and a human reference—on Point-Act. Success rates closely aligned with Point-Bench scores, yielding a strong linear correlation ($R^2 = 0.92$). This high correlation indicates that Point-Bench is a reliable proxy for the pointing capability of MLLMs in practical settings.

### 4.4 WHAT OTHER FACTORS DRIVE POINTING PERFORMANCE?

To understand the design choices that impact pointing, we conducted ablations on `GPT-4o` using variations in prompt structure and output representation, as shown in Table 1.

**Targeted prompts outperform verbose reasoning.** Incorporating Chain-of-Thought (CoT) reasoning reduced pointing accuracy by 2.9% for `GPT-4o` and by a substantial 16% for `Gemini-2.5-Flash`. Using raw, unfiltered user queries led to an additional drop of 2.6% and 3.7% for `GPT-4o` and `Gemini-2.5-Flash`, respectively. These results suggest that clear, targeted prompts with well-defined coordinate systems are crucial for effective pointing, while additional reasoning through language does not enhance MLLMs' pointing capabilities.

Table 1: Performance (%) of GPT-4o and Gemini-2.5-Flash variants across evaluation categories.

| Method | Affordance | Spatial | Reasoning | Steerability | Counting | Average |
|---|---|---|---|---|---|---|
| GPT-4o + In-Context (2-shots) | 46.0 | 26.7 | 23.9 | 22.5 | 33.2 | 30.4 |
| GPT-4o + Chain-of-Thought (CoT) (Wei et al., 2022) | 41.4 | 24.6 | 21.2 | 13.5 | 32.1 | 26.6 |
| GPT-4o + Unparsed language instruction | 37.8 | 23.7 | 19.7 | 21.9 | 31.1 | 26.9 |
| GPT-4o (Default) | 42.4 | 25.6 | 23.8 | 24.5 | 31.1 | 29.5 |
| Gemini-2.5-Flash + In-Context (2-shots) | 50.0 | 40.5 | 35.8 | 32.0 | 40.3 | 39.7 |
| Gemini-2.5-Flash + Chain-of-Thought (CoT) (Wei et al., 2022) | 44.4 | 25.6 | 24.9 | 26.0 | 33.7 | 30.9 |
| Gemini-2.5-Flash + Unparsed language instruction | 55.0 | 44.6 | 43.5 | 25.0 | 47.5 | 43.1 |
| Gemini-2.5-Flash (Default) | 58.6 | 47.7 | 43.2 | 35.0 | 49.7 | 46.8 |

## 5 LIMITATIONS, DISCUSSION, AND CONCLUSIONS

**Discussion.** PointArena evaluates spatial reasoning with a static benchmark (Point-Bench) and human preference battles (Point-Battle). We propose free-form contouring to replace grid tools for sharper masks, augment Point-Bench with fresh Point-Battle data, and use adaptive sampling to compare similarly strong models. **Limitations.** Annotations depend on SAM (Kirillov et al., 2023) and grid refinement, which can produce coarse masks on fine details. Static benchmarks risk leakage from public training data, and uniform Point-Battle sampling wastes comparisons between mismatched models. **Conclusion.** PointArena blends controlled tests with live feedback for scalable pointing evaluation. Our updates target better mask quality, fresher data, and more informative pairings—improving precision, scalability, and signal for real-world spatial reasoning.

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

## A APPENDIX

### A.1 POINT-ARENA EVALUATION

To evaluate each system, we consulted their official documentation to get their best performance. For models such as Gemini, Molmo, and Qwen, the documentation provided detailed guidelines on prompt construction for pointing tasks, including recommendations for coordinate systems. In these cases, we adhered to the official instructions. For other models, such as those from OpenAI, where explicit guidance was not available, we employed a standardized prompt format.

#### COORDINATES CHOICES

For most models, we adopted a standard XY coordinate system with the origin at the top-left corner. In this setup, the x-axis runs left to right, and the y-axis runs top to bottom. Models were instructed to return pixel-level coordinates directly in this format.

Gemini, however, uses a different convention: it outputs coordinates in [y, x] format with the same top-left origin, but normalized to a 0–1000 scale. To convert these outputs to pixel-level coordinates, we first divide by 1000, swap the x and y values, and then multiply by the actual image width and height.

Molmo models follow the same XY axis orientation as the standard setup but normalize coordinates to a 0–100 range. Thus, their outputs are converted by dividing by 100 and multiplying by the image's actual pixel-level width and height.

#### MODEL PROMPT TEMPLATES

The specific prompts used for each model are detailed below. For models that support separate system and user prompts, we present both components individually. For models that do not distinguish between system and user inputs, we provide a single unified prompt. Additionally, for certain models, the prompt used in the "counting" task category differs from that used in other categories. In these cases, we include both prompt versions to reflect the task-specific adaptations.

In the prompt templates, {object_name} refers to the refined user query, which always begins with the phrase "Point to." The placeholder {original_points_info} represents the coordinates of the original points when the task category is "Steerability." For all other categories, {original_points_info} is left as an empty string.

OPENAI MODELS

**System Content:**

>You are a helpful assistant that can identify objects in images and provide their coordinates.

**Prompt when `category == "counting"`:**

>{object_name}.
>The         image        dimensions         are         width={img_width}px,
>height={img_height}px.{original_points_info}
>The answer should follow the json format: [{"point": <point>}, ...].
>IMPORTANT: The points MUST be in [x, y] format where x is the horizontal position
>(left-to-right) and y is the vertical position (top-to-bottom) in PIXEL COORDINATES (not
>normalized).
>Example: For a point in the center of the image, return [width/2, height/2].

**Prompt otherwise:**

>{object_name}.
>The         image        dimensions         are         width={img_width}px,
>height={img_height}px.{original_points_info}
>The answer should follow the json format: [{"point": <point>}].
>IMPORTANT: Return EXACTLY ONE POINT. The point MUST be in [x, y] format where
>x is the horizontal position (left-to-right) and y is the vertical position (top-to-bottom) in
>PIXEL COORDINATES (not normalized).
>Example: For a point in the center of the image, return [width/2, height/2].

GEMINI MODELS

>{object_name}
>{original_points_info}

CLAUDE MODELS

**Prompt when `category == "counting"`:**

>You are a helpful assistant that can identify objects in images and provide their coordinates.

>{object_name}.
>The         image        dimensions         are         width={img_width}px,
>height={img_height}px.{original_points_info}
>The answer should follow the json format: [{"point": <point>}, ...].
>IMPORTANT: The points MUST be in [x, y] format where x is the horizontal position
>(left-to-right) and y is the vertical position (top-to-bottom) in PIXEL COORDINATES (not
>normalized).
>Example: For a point in the center of the image, return [width/2, height/2].

**Prompt otherwise:**

You are a helpful assistant that can identify objects in images and provide their coordinates.

{object_name}.
The image dimensions are width={img_width}px, height={img_height}px.{original_points_info}
The answer should follow the json format: [{"point": <point>}].
IMPORTANT: Return EXACTLY ONE POINT. The point MUST be in [x, y] format where x is the horizontal position (left-to-right) and y is the vertical position (top-to-bottom) in PIXEL COORDINATES (not normalized).
Example: For a point in the center of the image, return [width/2, height/2].

GROK MODELS

**Prompt when `category == "counting"`:**

You are a helpful assistant that can identify objects in images and provide their coordinates.

{object_name}.
The image dimensions are width={img_width}px, height={img_height}px.{original_points_info}
The answer should follow the json format: [{"point": <point>}, ...].
IMPORTANT: The points MUST be in [x, y] format where x is the horizontal position (left-to-right) and y is the vertical position (top-to-bottom) in PIXEL COORDINATES (not normalized).
Example: For a point in the center of the image, return [width/2, height/2].

**Prompt otherwise:**

You are a helpful assistant that can identify objects in images and provide their coordinates.

{object_name}.
The image dimensions are width={img_width}px, height={img_height}px.{original_points_info}
The answer should follow the json format: [{"point": <point>}].
IMPORTANT: Return EXACTLY ONE POINT. The point MUST be in [x, y] format where x is the horizontal position (left-to-right) and y is the vertical position (top-to-bottom) in PIXEL COORDINATES (not normalized).
Example: For a point in the center of the image, return [width/2, height/2].

QWEN MODELS

**System Content:**

You are a helpful assistant.

**Prompt:**

{object_name}
Output its coordinates in XML format <points x y>object</points>.
{original_points_info}

LLaVA MODELS

**System Content:**

You are a helpful assistant that can identify objects in images and provide their coordinates.

**Prompt when `category == "counting"`:**

{object_name}.
The image dimensions are width={img_width}px, height={img_height}px.{original_points_info}
For each point, give EXACT PIXEL COORDINATES in [x, y] format, where x is horizontal (left-to-right) and y is vertical (top-to-bottom).
Output format should be: [x, y], [x, y], etc. for multiple points.
ONLY return the coordinates with no additional text or explanations.

**Prompt otherwise:**

{object_name}.
The image dimensions are width={img_width}px, height={img_height}px.{original_points_info}
Give EXACT PIXEL COORDINATES in [x, y] format, where x is horizontal (left-to-right) and y is vertical (top-to-bottom).
ONLY return the coordinates with no additional text or explanations.

MOLMO MODELS

pointing: {object_name}
{original_points_info}

## A.2  POINT-BATTLE EVALUATION

For the Point-Battle evaluation, we employed an Elo rating system with a K-factor of 2. Given that we collected over 4,500 battle results, a lower K-factor was chosen to ensure greater rating stability. In each battle, two anonymous models were randomly selected from a pool of five: GPT-4o, Gemini 2.5 Flash Preview (04-17), Molmo-7B-D-0924, Qwen2.5-VL-7B-Instruct, and Grok-2 Vision (latest).

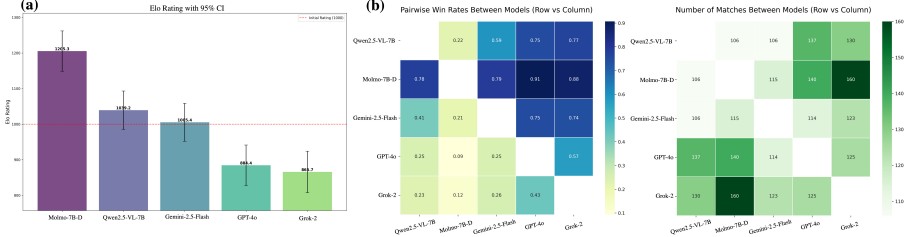

Figure 6: **Performance on Human Preference Evaluation with Point-Battle.** We collected over 4,500 votes from more than 100 global participants. Based on the Elo ratings derived from these votes, we observed a clear preference for outputs from open-source models such as `Molmo-7B-D` and `Qwen2.5-VL-7B`, which consistently outperformed proprietary models in terms of human preference.

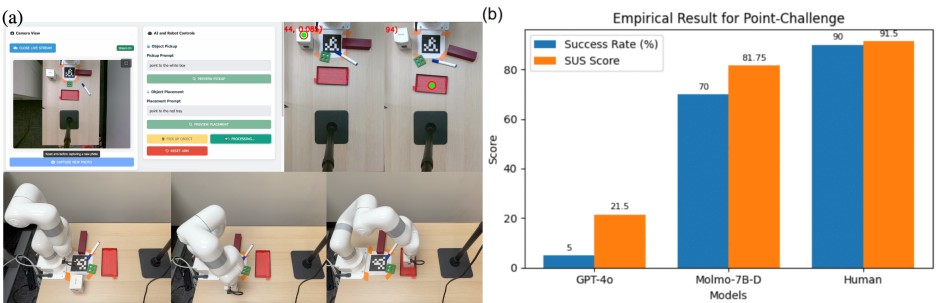

Figure 7: **Overview of the Point-Act system.** (a) The Point-Act manipulation setup enables remote control of a real-world xArm 6 Lite robot via language instructions, allowing users to evaluate pointing MLLMs. (b) User-blind evaluations and SUS preference scores collected for each model.

Affordance

Spatial

Counting

Reasoning

18

Steerable

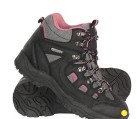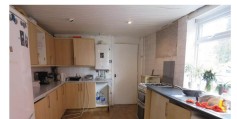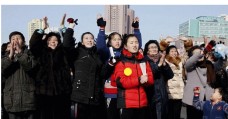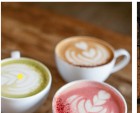