# OpenReview forum: "PointArena: Probing Multimodal Grounding Through Language-Guided Pointing"
_ICLR.cc/2026/Conference — ICLR 2026 Conference Withdrawn Submission_

### Official Review · Reviewer_FkoL · 2025-10-26

**Soundness:** 1
**Presentation:** 3
**Contribution:** 1
**Rating:** 2
**Confidence:** 2

**Summary:**

Authors introduced a benchmark for evaluating the image-langue-pointing capabilities of multi-model large language models. The benchmark devided the tasks into five categories based on a survey. This paper also evaluated various LLMs, providing valuable information for downstream users. Nevertheless, the benchmark seems to be an oversimplification of language condition segmentation, and the benchmark missed to evaluate "segment anything" baseline.

**Strengths:**

The paper proposed a benchmark on language-image-poinintng with clear definition. The benchmark extensively evaluated various LLMs. Authors also discovered that chain-of-thought decreases the performance on the point task.

**Weaknesses:**

My main concern is that the language-image-pointing task seems to be a simplification of language-image-segmentation task, where the LLM is prompted to segment out the object(s) of interest. One can subsample a pixel from the segmentation mask as the final output “point”. If my understanding is correct, then why do we need the pointing benchmark? Furthermore, why not include segment anything as a baseline? Authors can prompt the language question and subsample a pixel as a point.

**Questions:**

In addition to the main concern on the weakness section, I have following questions:

a. What are the implementation details of the point-act? How is the robot controlled?
b. In Figure 3, why “human”’s performance is not approaching 100% but is around 90%?
c. In Figure 4, in addition to colors, could authors use different markers for each LLMs for better interpretability?
d. Authors claim that “Pointing supervision significantly boosts performance.”, could you provide evidence that these model used “Point supervison”, and ideally provide an ablation study?
e. Authors claim that “Molmo excels on Point-Act evaluation.”, could you explain why Molmo outperformed other baselines? Is it due to Molmo's architecture or training pipeline?

---

### Official Review · Reviewer_deBG · 2025-10-31

**Soundness:** 3
**Presentation:** 2
**Contribution:** 3
**Rating:** 6
**Confidence:** 4

**Summary:**

This work proposes Point-Bench, a curated dataset comprising approximately 1,000 pointing tasks across five reasoning categories, and evaluates both closed-source and open-source models on it. It further introduces Point-Battle, a framework for assessing VLMs’ pointing capabilities based on human preferences. Finally, the study demonstrates that Point-Bench accuracy strongly correlates with real-world manipulation task success.

**Strengths:**

The overall writing is clear and easy to follow.

The paper presents a well-motivated study that focuses on benchmarking the pointing ability of VLMs, with a clearly articulated purpose.

In addition to reporting standard success rates, it introduces Point-Battle and Point-Act as complementary evaluation metrics, providing a more comprehensive assessment.

**Weaknesses:**

The paper lacks detailed statistics of the benchmark, such as the types of scenarios and object categories included, which are essential for understanding its coverage and diversity.

Although Point-Act is presented as a key contribution, the paper provides insufficient details about the specific tasks performed by the robot and how the pointing ability translates into or supports manipulation performance.

The inference time of different models is not reported. Moreover, as a benchmarking paper, it would be beneficial to include information on the cost and feasibility of running the benchmark to facilitate future evaluations by other researchers.

**Questions:**

What types of tasks are evaluated in Point-Act, and how are manipulation actions derived from the model’s pointing outputs? I would assume the use of a depth image or 3D information, but this part remains unclear.

Is there a way to develop a more precise metric for measuring pointing ability? Using “point within the mask” seems to be only a rough criterion.

How does the model perform in terms of cross-view pointing ability i.e., consistency of pointing performance across different camera perspectives?

If different forms of prompts are used to query the MLLMs, would the evaluation results on this benchmark vary? Is it possible to provide additional experimental results to further verify the consistency and robustness of the benchmark?

---

### Official Review · Reviewer_cWoh · 2025-11-01

**Soundness:** 3
**Presentation:** 2
**Contribution:** 3
**Rating:** 4
**Confidence:** 4

**Summary:**

This paper proposed PointArena, a comprehensive evaluation platform for multimodal pointing evaluation.
It consists of a dataset with 5 reasoning categories (Point-Bench), an interactive web-based arena (Point-Battle), and a real-world robot manipulation system (Point-Act).
The authors also conducted experiments with existing SoTA models and discussed insights about the role of pointing in real world reasoning-based tasks.

**Strengths:**

The dataset consists of 5 representative and challenging reasoning/localization tasks, making this dataset able to evaluate various aspects of multimodal point comprehension. Point-Battle is also a good approach to address the lack of referral image-text pairs and accurate evaluators.

**Weaknesses:**

1. Point-Act is under-explained. There's too little information about how the robot and experiments are set up. Evaluation metrics, examples, results, and how to access this platform, are missing. As robot pick-and-place is not only related to pointing itself, it's essential to clarify the rest of the setting, such as how the grasp motion and controller are implemented, and therefore how this platform contributes to the entire PointArena system.
2. Sec 5 seems unfinished. Discussion, limitation, and conclusion look like summarizations of Sec 3, rather than drawing any insights from the experiments.
3. Appendix is messy. Please make them more organized and clearly explained, rather than listing prompts and unscaled figures randomly, especially when you have some of your main results in the Appendix.

**Questions:**

Point-Bench consists of 982 image-question pairs. Considering the diversity of open-world combinations of objects and relationships, is this scale sufficient to draw conclusions about model performance, especially when divided into 5 categories? As you said, it is totally possible to have leakage when evaluating up-to-date VLMs. Is integrating data from Point-Battle in the future your solution to this problem?

---

### Official Review · Reviewer_CNEY · 2025-11-01

**Soundness:** 3
**Presentation:** 3
**Contribution:** 2
**Rating:** 6
**Confidence:** 4

**Summary:**

The paper introduces PointArena, a three-stage evaluation suite for language-guided pointing with multimodal models: Point-Bench (a curated static benchmark of 982 image–query pairs across five reasoning categories), Point-Battle (a live, blinded, head-to-head arena that has collected >4.5k votes from around 100 participants), and Point-Act (a real-world robotic setup where models' predicted points drive an xArm to execute pick/place). The task is formalized as predicting one or more image-space coordinates that must land inside ground-truth masks, whose success is measured via a binary mask-coverage criterion with standardized [x, y] outputs. Evaluations of both proprietary and open-source MLLMs indicate Molmo-72B attains the highest Point-Bench accuracy, though some proprietary models are close; pointing-specific supervision yields large gains; and Point-Bench accuracy correlates strongly with Point-Battle preferences and Point-Act success. Ablations suggest targeted prompts beat chain-of-thought for pointing.

**Strengths:**

1. The paper is well written and easy to follow.
2. Treating pointing as a precise grounding interface is timely for robotics, assistive tech, and interactive systems.
3. Point-Bench covers five complementary categories (Spatial, Affordance, Counting, Steerable, Reasoning), enabling diagnostic analyses beyond simple referring localization. Point-Battle brings scalable preference data with blinded pairwise comparisons and anonymized voting. Point-Act grounds claims in physical manipulation on an xArm 6 Lite, directly testing whether pointing precision transfers to real tasks.
4. Strong cross-stage correlations between Point-Bench and Point-Battle (R²≈0.85), and Bench and Act (R²≈0.92), strengthen external validity.

**Weaknesses:**

1. For non-counting tasks only the first returned point is scored, which may bias against models returning ranked or structured outputs. Consider top-k or confidence-weighted evaluation.
2. For counting or multi-target cases, the success criterion is set coverage without precision/recall trade-offs. A bipartite matching metric (Hungarian assignment with distance thresholds) plus F1 would better reflect partial correctness and over-/under-pointing behavior.
3. The Arena uses Elo with K=2 for stability. However, there is little discussion of voter quality control, duplicate participant mitigation, regional/task mixture balance, or defenses against strategic voting.
4. Point-Act reports relative statements (e.g., substantial margin) but omits granular success definitions (reach/pick/place), latency, and failure modes (depth errors, gripper slip, calibration).

**Questions:**

1. I recommend to include a consolidated results table: per-category and average accuracy for all 16 models, with mean±SD over 3 runs and 95% CIs and annotate statistically indistinguishable groups.
2. For the R²=0.85/0.92 findings, it would be better to provide scatter plots, model counts, bootstrapped CIs, and sensitivity checks (e.g., exclude models trained on pointing data; remove one family at a time).
3. It would be better to report inter-annotator agreement on masks (e.g., mean IoU across verifiers) and a boundary-sensitivity study (success vs. distance to mask edges). Given SAM dependence, can you estimate error rates on thin structures? A small expert-reannotated subset could quantify bias.
4. Did you run near-duplicate checks vs. PixMo/RoboPoint/RefCOCO family? Any overlap screening against common pretraining sets (e.g., LAION)? Since you argue pointing supervision drives gains, leakage audits will make the causal story more persuasive.
5. Why CoT Hurts Pointing? Can you analyze failure modes (e.g., verbose reasoning truncating coordinates, distractor attention, incorrect normalization)?

---

### Note · Authors · 2025-11-16

I have read and agree with the venue's withdrawal policy on behalf of myself and my co-authors.